# Biochemistry and Diseases Related to the Interconversion of Phosphatidylcholine, Phosphatidylethanolamine, and Phosphatidylserine

**DOI:** 10.3390/ijms251910745

**Published:** 2024-10-06

**Authors:** Jan Korbecki, Mateusz Bosiacki, Patrycja Kupnicka, Katarzyna Barczak, Paweł Ziętek, Dariusz Chlubek, Irena Baranowska-Bosiacka

**Affiliations:** 1Department of Anatomy and Histology, Collegium Medicum, University of Zielona Góra, Zyty 28, 65-046 Zielona Góra, Poland; jan.korbecki@onet.eu; 2Department of Biochemistry and Medical Chemistry, Pomeranian Medical University, Powstańców Wlkp. 72, 70-111 Szczecin, Poland; mateusz.bosiacki@pum.edu.pl (M.B.); patrycja.kupnicka@pum.edu.pl (P.K.); dchlubek@pum.edu.pl (D.C.); 3Department of Conservative Dentistry and Endodontics, Pomeranian Medical University, Powstańców Wlkp. 72, 70-111 Szczecin, Poland; katarzyna.barczak@pum.edu.pl; 4Department of Orthopaedics, Traumatology and Orthopaedic Oncology, Pomeranian Medical University, Unii Lubelskiej 1, 71-252 Szczecin, Poland; pawel.zietek@pum.edu.pl

**Keywords:** phospholipid, phosphatidylethanolamine N-methyltransferase, phosphatidylserine decarboxylase, phosphatidylcholine

## Abstract

Phospholipids are crucial structural components of cells. Phosphatidylcholine and phosphatidylethanolamine (both synthesized via the Kennedy pathway) and phosphatidylserine undergo interconversion. The dysregulation of this process is implicated in various diseases. This paper discusses the role of enzymes involved in the interconversion of phosphatidylcholine, phosphatidylethanolamine, and phosphatidylserine, specifically phosphatidylethanolamine N-methyltransferase (PEMT), phosphatidylserine synthases (PTDSS1 and PTDSS2), and phosphatidylserine decarboxylase (PISD), with a focus on their biochemical properties. Additionally, we describe the effects of the deregulation of these enzymes and their roles in both oncological and non-oncological diseases, including nonalcoholic fatty liver disease (NAFLD), Alzheimer’s disease, obesity, insulin resistance, and type II diabetes. Current knowledge on inhibitors of these enzymes as potential therapeutic agents is also reviewed, although in most cases, inhibitors are yet to be developed. The final section of this article presents a bioinformatic analysis using the GEPIA portal to explore the significance of these enzymes in cancer processes.

## 1. Introduction

Phospholipids, essential components of cellular lipid membranes, can be categorized based on their structure, specifically the type of polyhydric alcohol forming the phospholipid (glycerol [1] or sphingosine [2]) and the compound attached to the phosphate group (choline, ethanolamine, serine, glycerol, inositol) [1]. Phospholipids also incorporate various fatty acids, which can modify their properties and influence the function of cell membranes [3,4]. Both the cell membrane and intracellular membranes are composed of two leaflets. In these layers, phospholipids are arranged with their fatty acid chains facing the interior of the membrane. Additionally, cell membranes contain other molecules, such as cholesterol and proteins. The phospholipid composition of cell membranes significantly influences the properties of these cell structures and, consequently, it is tightly regulated. The primary mechanism for modifying lipid membrane composition is the regulation of the synthesis of individual phospholipids. The composition of cell membrane leaflets is further regulated by enzymes known as flippases, which facilitate the transfer of phospholipids between the layers of the membrane [5].

The primary de novo synthesis pathway for phosphatidylcholine (PC) and phosphatidylethanolamine (PE) is the Kennedy pathway [6]. This pathway consists of three stages. First, choline or ethanolamine is phosphorylated to form phosphocholine and phosphoethanolamine, respectively. These intermediates are then converted into cytidine diphosphate (CDP)-choline and CDP-ethanolamine. In the third and final step of the Kennedy pathway, these intermediates are combined with diacylglycerol (DAG) to produce PC and PE, respectively.

The Kennedy pathway does not synthesize other phospholipids; these are produced via alternative pathways. For instance, phosphatidylserine (PS) is synthesized from PC or PE through the exchange of choline or ethanolamine with serine. The enzymes catalyzing this reaction are phosphatidylserine synthase (PTDSS)1 and PTDSS2 [7,8]. Phospholipid composition can also be modified through other phospholipid conversion reactions (Figure 1). An example is the methylation of ethanolamine in PE by phosphatidylethanolamine N-methyltransferase (PEMT), resulting in the formation of PC from PE [9,10,11]. Additionally, serine in PS can undergo decarboxylation to produce PE, catalyzed by phosphatidylserine decarboxylase (PISD/PSD) [12].

Phospholipid metabolism undergoes alterations in numerous diseases. However, there is no comprehensive overview detailing phospholipid transformations in these pathological conditions. To address this knowledge gap, the present review consolidates all available information concerning the mutual interconversions of PC, PE, and PS under physiological conditions and in key pathological states. Furthermore, the review aims to elucidate potential therapeutic strategies targeting the metabolism of these phospholipids for the treatment of such diseases.

## 2. Phosphatidylethanolamine N-Methyltransferase

### 2.1. Biosynthesis of Phosphatidylcholine Involving Phosphatidylethanolamine N-Methyltransferase

PC can be synthesized from PE. The enzyme responsible for catalyzing this reaction is PEMT [9,10,11,13], the only enzyme responsible for the methylation pathway of PE to form PC and catalyzing all three methylation reactions. The *PEMT* gene in humans produces three transcripts that differ in their 5′-UTR and encode PEMT proteins of different sizes [14,15]. Transcript 1 generates a longer isoform of PEMT (PEMT-L), while transcripts 2 and 3 produce a smaller isoform of PEMT (PEMT-S) [16]. Both isoforms differ at the N-terminus, with PEMT-L being 36 amino acids longer than PEMT-S. Unlike PEMT-S, PEMT-L is N-glycosylated at Asn13 with high-mannose oligosaccharides at the N-terminus. PEMT-S shows higher activity than PEMT-L, and the two isoforms exhibit different substrate specificity. PEMT-S is involved in the synthesis of PC species with long polyunsaturated acyl chains [16].

The tissue-specific expression of each of the three PEMT transcripts varies, with the liver expressing all three transcripts. These transcripts arise from alternative splicing and possibly from alternative transcription start sites. Through PEMT, PC containing higher levels of polyunsaturated fatty acids (PUFA) is produced compared to what occurs in the Kennedy pathway [9,10,17].

PEMT is an enzyme that transfers three methyl groups from S-adenosyl-L-methionine to convert ethanolamine to choline in PE [9,10,11]. It is located in the endoplasmic reticulum, mitochondria-associated membranes [18], and the nucleus [19]. This enzyme is crucial for the production of PC and in the formation of lipid droplets, organelles that are surrounded by glycerophospholipids mainly composed of PC [20].

The liver exhibits the highest expression of PEMT, while lower expression is observed in muscles, kidneys, brain, heart, and adipose tissue [21]. PEMT plays a pivotal role in synthesizing PC in the liver, serving as a crucial pathway for choline production in this organ [22,23,24].

Furthermore, due to its involvement in the metabolism of phospholipids containing PUFA, PEMT contributes to PUFA homeostasis throughout the body. Experimental evidence indicates that a defect in the *PEMT* gene leads to reduced PUFA levels in the serum [25]. Additionally, specific polymorphisms in the *PEMT* gene are associated with a decline in PUFA levels in the bloodstream [26].

PEMT expression is also discernible in brown adipose tissue (BAT), where it orchestrates the overall metabolism [27]. Moreover, PEMT is indispensable for the expression of uncoupling protein 1 (UCP1) in BAT, thereby influencing thermogenesis [28]. Despite these crucial roles, the precise mechanism by which PEMT impacts UCP1 remains unknown, especially since it does not directly correlate with PEMT expression in adipose tissue, liver, or muscles.

### 2.2. Phosphatidylethanolamine N-Methyltransferase in Cancers

PEMT’s role in oncogenic processes remains inadequately explored, with only several experimental studies available.

PEMT exhibits dual pro- and anti-tumor properties, dependent on the specific cancer type. In hepatocellular carcinomas, PEMT displays anti-oncogenic characteristics [29,30]. Tumor tissues show a lower PEMT expression compared to healthy liver tissue, and a heightened PEMT expression in hepatocellular carcinomas correlates with a more favorable prognosis. This association is linked to DNA methylation, as PEMT utilizes S-adenosyl-L-methionine to modulate DNA methyltransferase-catalyzed reactions, influencing the availability of this substance and consequent changes in DNA methylation patterns. Consequently, diminished PEMT expression in liver cells leads to reduced expression of F-box only protein 31 (Fbxo31) and hepatocyte nuclear factor 4α (HNF4α), thereby increasing cyclin D1 activity and hepatocyte proliferation [19]. Thus, decreased PEMT expression acts as a pro-oncogenic factor in liver tumors.

PEMT demonstrates pro-oncogenic properties in diverse tumor types. Colon adenocarcinoma witnesses an increase in PEMT expression, accompanied by a reduction in choline kinase β (CHKβ) expression [31]. This suggests a decrease in PC production through the Kennedy pathway in this tumor, coupled with an increase facilitated by PEMT. Another study reveals heightened PEMT expression in non-small-cell lung cancer (NSCLC) tumors compared to surrounding tissue [32]. Elevated gene expression correlates with poorer prognoses for NSCLC patients [32].

The precise pro-oncogenic impact of PEMT remains unknown. The observed effect may be linked to PC and choline production, as elevated blood choline levels are associated with a heightened susceptibility to cancer [33]. During tumor transformation, cells increase their demand for choline [34], and heightened PEMT activity might satisfy this demand in specific tumor types. PEMT also participates in the one-carbon metabolism pathway, utilizing S-adenosyl-L-methionine for PE methylation. Consequently, PEMT competes for this substrate with other methyltransferases [35]. High PEMT activity reduces DNA methyltransferase activity, altering gene expression and potentially leading to tumorigenesis. Another pro-oncogenic aspect of PEMT is its influence on p53. PEMT can bind to clathrin heavy chain and p53 [19], diminishing p53’s transcriptional activity. Given that p53 is a protein inhibiting tumorigenesis [36], increased PEMT expression might facilitate tumor development.

A polymorphism in the SNP rs12325817 within the *PEMT* gene is linked to an increased risk of developing breast cancer [37]. Additionally, the hypermethylation of the -132 site in the *PEMT* promoter occurs in breast cancer type 1 susceptibility gene (*BRCA1*)-mutated breast cancer cells [38], resulting in reduced enzyme expression. However, PEMT gene methylation does not correlate with prognoses in patients with BRCA1-mutated breast cancer [38]. Furthermore, specific *PEMT* single nucleotide polymorphisms (SNPs), notably rs2124344, rs7215833, rs4646340, rs4646350, and rs4646341, are associated with an elevated risk of developing bladder cancer [35]. This association may be connected to alterations in DNA methylation.

The critical role of PEMT in oncogenic processes suggests it is a potential therapeutic target. Specifically, an agent targeting PEMT may demonstrate efficacy against NSCLC, although the therapeutic outcome will depend on the cancer type. For example, in hepatocellular carcinomas, PEMT functions as a tumor suppressor, and thus such an agent would lack therapeutic effectiveness.

### 2.3. Phosphatidylethanolamine N-Methyltransferase in Metabolic and Hepatic Diseases

PEMT plays a pivotal role in the context of obesity. Studies on polymorphisms have revealed that the SNP rs4646404 and SNP rs4646343 in the *PEMT* gene are linked to an increased risk of obesity [39]. Additionally, the G genotype of SNP rs7946 in *PEMT*, in comparison to the A genotype, is associated with an elevated body mass index (BMI) [40]. There exists a positive correlation between the expression of PEMT in adipose tissue and both the BMI and waist-to-hip ratio [39,41].

Investigations involving mice with a defective *PEMT* gene have affirmed the indispensable role of this enzyme in obesity development [23]. Animal studies have shed light on its connection with PC production, a crucial source of choline, particularly in the liver [22,23,24]. However, not all consequences of reduced PEMT expression and activity can be attributed solely to decreased choline production in the liver [25].

Diminished choline levels lead to a reduction in stearoyl-CoA desaturase (SCD) expression, decreased monounsaturated fatty acids (MUFA) production in the liver, and heightened fatty acid oxidation. Notably, decreased PEMT activity results in lower PC levels in mitochondria, subsequently boosting mitochondrial respiration [42].

PEMT in the liver is also accountable for insulin resistance in obesity, as evidenced by experiments on animals subjected to a high-fat diet [24,43]. Reduced PEMT activity in the liver shields against insulin resistance, linked to a decrease in the PC:PE ratio [43]. This is linked to the activation of the aryl hydrocarbon receptor (AhR) by PC [44], which reduces the expression of insulin receptor substrate-2 (IRS-2) and, consequently, impairs signal transduction from the insulin receptor.

PEMT’s impact on obesity may also be contingent on lipogenesis in white adipose tissue [45]. Studies involving animals fed a high-fat diet have indicated that PEMT activity is imperative in lipogenesis. Interestingly, PEMT does not influence muscle metabolism, specifically β-oxidation [46]. However, PEMT may indirectly affect mitochondrial function in muscles by enhancing sarco/endoplasmic reticulum Ca^2+^-ATPase Ca^2+^ transport efficiency [47].

The consequences of reduced PEMT expression and activity, such as the absence of obesity in animals fed a high-fat diet [23,48], underscore the pivotal role of PEMT. Moreover, it is indispensable in adipocyte differentiation [49]. Nonetheless, studies on animals have not definitively established a direct link between adipocyte differentiation and PEMT in obesity [45].

PEMT is intricately associated with nonalcoholic fatty liver disease (NAFLD), particularly nonalcoholic steatohepatitis (NASH) [19,50]. The SNP rs7946 (+5465G → A) in PEMT, encoding a substitution of valine with methionine at position 175 (V175M) in PEMT, is linked to an augmented risk of developing NAFLD, including NASH, especially in non-obese East Asian individuals [50,51,52,53]. The risk of NAFLD with this genotype is associated with insufficient choline intake [40]. Additionally, one study suggested that this association may be gender-specific [40]. SNP rs7946 (+5465G → A) PEMT/V175M PEMT is associated with reduced PEMT activity [50], indicating that PEMT serves as a protective factor against NASH development.

Studies involving animals and humans have consistently demonstrated lower PEMT expression levels in the livers of individuals with NASH, decreasing with the degree of fibrosis [54]. In adipose tissue, PEMT expression is higher in patients with NASH than in healthy individuals [41]. Animal studies have shown that reducing PEMT activity in the liver results in increased triacylglycerol (TAG) accumulation, a higher number and size of lipid droplets, inflammatory reactions, and liver fibrosis [23,24,43,45]. This is intricately tied to a disturbance in the PC to PE ratio in the liver, leading to endoplasmic reticulum stress.

Moreover, decreased PEMT expression induces changes in DNA methylation, causing a reduction in the expression of Fbxo31 and HNF4α, amplifying cyclin D1 activity, and, consequently, hepatocyte proliferation [19]. Perturbations in PEMT activity lead to liver cell apoptosis, as demonstrated in experiments with mice with a *PEMT* gene defect [19]. Consequently, decreased PEMT activity in the liver increases the risk of developing NASH.

The data presented indicate that PEMT inhibitors may be agents for treating or preventing obesity. However, they may also have side effects, including the development of NASH.

PEMT also contributes to steatosis in the course of chronic hepatitis C virus (HCV) infection [55]. HCV increases PEMT expression in hepatocytes, a process dependent on the HCV genotype. Increased PEMT expression enhances lipid accumulation and supports HCV replication, leading to disease progression and liver steatosis. This suggests that PEMT inhibitors may be potential drugs for inhibiting HCV replication.

### 2.4. Phosphatidylethanolamine N-Methyltransferase in Type II Diabetes

PEMT plays a significant role in type II diabetes, as substantiated by the correlation observed between the SNP rs4646404 in the *PEMT* gene and insulin resistance in individuals with obesity [41]. Furthermore, the expression of PEMT in adipose tissue demonstrates a positive correlation with insulin resistance in those with obesity [41]. Additionally, PEMT activity escalates in the liver under diabetic conditions, as evidenced by studies conducted on rats treated with streptozotocin [56].

The heightened expression of PEMT in the liver leads to an increase in PC levels and a decrease in PE levels in this organ. The resulting elevated PC:PE ratio in the liver contributes to hepatic insulin resistance [43]. This is related to the activation of AhR by PC, which decreases the expression of IRS-2 in the liver [44]. This reduction impairs signal transduction from the insulin receptor, leading to insulin resistance. Investigations on rats treated with streptozotocin also indicate the involvement of PEMT in diabetic nephropathy [21].

These data suggest that the use of a specific inhibitor targeting PEMT holds the potential to enhance insulin effectiveness in the liver and mitigate renal complications arising from the detrimental effects of diabetes.

### 2.5. Phosphatidylethanolamine N-Methyltransferase in Atherosclerosis

The discussed enzyme emerges as a promising therapeutic target in the fight against atherosclerosis development. PEMT’s pivotal role in liver lipid metabolism positions it as a crucial source of PC and, consequently, choline [22,23,24]. The decreased expression of this enzyme in the liver results in reduced levels of very low-density lipoprotein (VLDL), low-density lipoprotein (LDL), TAG, and cholesterol in the bloodstream, as evidenced by experiments conducted on mice subjected to a high-fat diet [57,58,59]. This is linked to the involvement of PC in lipoprotein assembly and secretion [60]. Consequently, these findings suggest that medications acting as PEMT inhibitors hold promise for use in the treatment or prevention of atherosclerosis.

### 2.6. Phosphatidylethanolamine N-Methyltransferase in Neurological Diseases

PEMT is an essential enzyme for the proper functioning of the brain. Studies on mice with a *PEMT* gene defect revealed significant differences in the expression of various genes in the brain [61]. Specifically, PEMT reduces the proliferation of progenitor cells [62] but plays a crucial role in neural cell differentiation [61]. Consequently, changes in the activity of this enzyme may lead to certain neurological disorders. An example of this is schizophrenia, where the susceptibility to schizophrenia has been linked to the SNP rs464396 in the *PEMT* gene [63]. However, there are also studies that do not confirm that this polymorphism in the *PEMT* gene is associated with schizophrenia [64].

PEMT may be significant in Alzheimer’s disease. It has been shown that the AA genotype of the SNP rs7949 in the *PEMT* gene is associated with an increased likelihood of developing Alzheimer’s disease [65]. In this disease, PEMT activity is also lower in the frontal cortex than in healthy individuals of the same age [66]. This may explain the altered levels of various phospholipids in the brains of people with Alzheimer’s disease, particularly the reduction in plasmenylcholine containing docosahexaenoic acid (DHA) [67,68]. The decrease in plasmenylcholine levels may enhance the accumulation of β-amyloid (Aβ) plaques in the brains of individuals with Alzheimer’s disease, thereby increasing the progression of the disease [69].

### 2.7. Phosphatidylethanolamine N-Methyltransferase in Ontogeny

There is evidence suggesting that PEMT plays a crucial role in human embryonic development. A study conducted in the Chilean population has identified associations between the SNPs rs7649 and rs4646409 in the *PEMT* gene and nonsyndromic cleft lip with or without cleft palate [70]. Furthermore, the AA genotype of the SNP rs7949 in mothers has been linked to an increased likelihood of preterm birth, especially in cases of choline deficiency [71].

The discussed enzyme, with the genotype SNP rs7946 (+5465G → A) *PEMT*/V175M PEMT, exhibits reduced activity [50], providing an explanation for the abnormal course of pregnancy with low choline intake in women with this genotype. A Polish study has also uncovered an association between intrauterine fetal death and SNPs rs4646406, rs4244593, and rs8974 in the *PEMT* gene in women [72]. These findings highlight the essential role of choline and its metabolism in the proper development of the human embryo during prenatal life.

### 2.8. Inhibitors of Phosphatidylethanolamine N-Methyltransferase

Currently, research on PEMT inhibitors is limited, primarily due to the lack of knowledge regarding specific inhibitors. The inhibitors being tested have a structural similarity to the substrate for PEMT, S-adenosyl-L-methionine. Compounds such as 3-deazaadenosine [73,74] and aminoimidazole-4-carboxamide ribonucleotide [75] have been studied as potential inhibitors of this enzyme (Figure 2). However, these compounds, classified as S-adenosylhomocysteine hydrolase inhibitors, demonstrate limited specificity toward the targeted enzyme [76,77]. As a result, these compounds have not undergone clinical testing. PEMT plays a role in several significant diseases, including obesity, obesity-related insulin resistance, HCV infection, type II diabetes, and atherosclerosis. Therefore, the development and clinical testing of specific PEMT inhibitors are necessary.

## 3. Biosynthesis of Phosphatidylserine

### 3.1. Phosphatidylserine Synthase

After the biosynthesis of PC and PE, these glycerophospholipids can be converted to PS through the action of phosphatidylserine synthases: PTDSS1 [7] and PTDSS2 [7,8]. PTDSS1 and PTDSS2 are located in the mitochondria-associated membranes [78] and endoplasmic reticulum [79,80].

PTDSS1 shows substrate specificity to PC, which has stearic acid C18:0 at the sn-1 position and DHA C22:6n-3 at the sn-2 position [81]. The activity of this enzyme is not inhibited by its own product, PS [7]. PTDSS1 converts PE to PS and PC to PS and catalyzes the reverse reaction [7].

PTDSS2 converts PE to PS and catalyzes the reverse reaction [7,82]. However, this enzyme also shows some activity towards PE plasmalogens, indicating its possible involvement in ether lipid biosynthesis [82]. PTDSS2 prefers DHA C22:6n-3 at the sn-2 position in PE but does not show substrate preference for any particular fatty acid at the sn-1 position in PE. The activity of PTDSS2 is inhibited by the product of the catalyzed reaction, i.e., PS [7].

Gain-of-function mutations in the *PTDSS1* gene result in the increased production of PS, which leads to Lenz–Majewski hyperostotic dwarfism [83,84]. Conversely, loss-of-function mutations in the *PTDSS1* gene lead to developmental delays [85].

### 3.2. Phosphatidylserine Synthases in Neurological Diseases

PTDSS1 and PTDSS2 are significant in the progression of certain diseases, particularly in brain function. This is associated with the higher concentration of PS in the brain compared to other tissues [86]. In the brain, a substantial proportion of PS incorporates DHA in its structure [86,87].

PTDSS plays a crucial role in synaptogenesis and axonal growth [88]. Both these enzymes are also vital for the proper functioning of mitochondria in neural tissue. Disruptions in the activity of these enzymes can lead to impaired brain function. Loss-of-function mutations in the *PTDSS1* gene have been identified, correlating with some cases of developmental delay [85]. Some studies suggest that this gene may be relevant in the development of Autism Spectrum Disorders (ASD) [89]. Additionally, research on the cingulate cortex and prefrontal cortex indicates lower PTDSS2 expression in individuals with major depressive disorder compared to healthy individuals [90]. This implies that lipid metabolism is crucial for proper brain function, and alterations in this metabolism can lead to neurological disorders.

### 3.3. Phosphatidylserine Synthases in Metabolic Diseases

Phosphatidylserine synthases play a role in metabolic diseases. In adipose tissue, there is a positive correlation between PTDSS2 expression and the BMI and waist-to-hip ratio [39]. However, the significance of this enzyme in obesity is not fully understood. It is suspected that PTDSS2 and PS may be related to obesity by influencing thermogenesis [91] or another process that needs further exploration for a better understanding of obesity.

### 3.4. Phosphatidylserine Synthases in Myocardial Infarction

Phosphatidylserine synthases may be relevant in myocardial infarction. During myocardial infarction, there is a reduction in PTDSS1 expression in heart cells [92]. This is associated with oxygen–glucose deprivation in heart cells during a heart attack. Consequently, a decrease in PS levels in heart tissue occurs. Deficiencies in PS in cardiomyocytes lead to the apoptosis of these cells after a heart attack [92]. This opens up a potential therapeutic pathway, where supplementing PS for individuals immediately after a heart attack may protect the heart from further damage. On the other hand, a study on mice has shown that PS supplementation after myocardial infarction may not have a therapeutic effect [93]. However, PS supplementation prior to infarction provides protection against the consequences of myocardial infarction, which is an effect linked to the increased expression of protein kinase C-ε (PKC-ε).

### 3.5. Phosphatidylserine Synthases in Cancer

PTDSS1 and PTDSS2 are involved in cancer processes. Apoptosis is a characteristic process in cancer and drives tumor development [94]. PS in the cell is located in the inner leaflet of the cell membrane. During apoptosis, PS appears on the outer leaflet of the cell membrane [95], which is related to the blocking of the flippases responsible for the asymmetry in the composition of phospholipids between the cell membrane layers. PS on the outer leaflet of the cell membrane acts as an “eat me” signal for macrophages to remove apoptotic bodies [96]. However, this process serves as a pro-tumorigenic mechanism. PS exposed by cells in an apoptotic state acts on tumor-associated macrophages (TAM) [96,97], enhancing their proliferation and promoting their M2 polarization. This represents a macrophage phenotype that supports tumor development.

Non-apoptotic PS on cancer cells can significantly impact the tumor microenvironment. It appears on the outer leaflet of the cell membrane when flippase activity decreases, which is particularly evident in breast cancer. In cases of *BRCA1* gene defects, there is a reduction in flippase ATP11b expression and an increase in PTDSS2 expression. Consequently, non-apoptotic PS accumulates on the outer leaflet of the cell membrane, leading to the accumulation of immunosuppressive cells, such as myeloid-derived suppressor cells (MDSC), which promote tumor development, especially by enhancing the release of cancer cells into the blood and promoting metastasis [98].

The significance of phosphatidylserine synthases has been demonstrated in specific types of cancer. In hepatocellular carcinoma, higher PTDSS2 expression in the tumor is associated with a poorer prognosis [99]. On the other hand, in breast cancer, the level of PTDSS2 expression does not impact prognosis [98,100]. When considering the expression of two genes, *ATP11B* and *PTDSS2*, simultaneous low *ATP11B* expression and higher *PTDSS2* expression are associated with worse prognoses compared to patients with higher *ATP11B* and lower *PTDSS2* expression in the tumor [98].

### 3.6. Phosphatidylserine Synthase Inhibitors as Anticancer Drugs

Specific inhibitors for PTDSS1 have already been developed, such as DS07551382, DS55980254 [101], and DS68591889 [102] (Figure 3). In vitro and in vivo studies have shown that these inhibitors are effective against tumors with a deletion in the tumor-suppressive locus 11p15.5, where the *PTDSS2* gene is located, representing the second isoform of phosphatidylserine synthase [101]. These inhibitors also demonstrate efficacy against tumors in which PS metabolism plays a significant role in cancer processes, for example, B cell lymphomas [102]. Blocking PTDSS1 activity in these tumors deregulates phosphatidylinositol-4-monophosphate (PI4P) metabolism, affecting B cell receptor (BCR) function, ultimately leading to cancer cell apoptosis. However, these inhibitors have not yet been clinically tested. They hold potential as anticancer agents for tumors with a deletion in 11p15.5.

## 4. Biosynthesis of Phosphatidylethanolamine Involving Phosphatidylserine Decarboxylase

### 4.1. Phosphatidylserine Decarboxylase

PE can be synthesized from PS in a reaction catalyzed by PISD/PSD [12]. An isoform of PISD produced by alternative splicing has also been described. PISD has been found in mitochondria [103,104,105] and on lipid droplets, and is important for the functioning of these organelles [106]. These are not the only differences between the two pathways of PE synthesis. PE that has PUFA, especially DHA, at the sn-2 position is preferentially produced with the involvement of PISD [107,108]. Meanwhile, the Kennedy pathway preferentially produces PE with MUFA at the sn-2 position and PUFA with only two double bonds in the hydrocarbon chain.

PISD plays a vital physiological role, particularly in muscle physiology [109]. A decrease in the activity of this enzyme in muscles results in severe myopathy. Additionally, PISD is expressed in the brain [110], but its activity in the brain diminishes with age, as evidenced by experiments on rats. This decline may influence alterations in lipid metabolism in older individuals, consequently contributing to age-related brain diseases. Furthermore, PISD is indispensable in human development, and mutations in the *PISD* gene lead to spondyloepimetaphyseal dysplasia [111].

### 4.2. Phosphatidylserine Decarboxylase in Cancer

PISD plays a notable role in cancer processes. During the epithelial–mesenchymal transition (EMT) of cancer cells, there is a reduction in the expression of this enzyme [112]. This is significant as it facilitates subsequent tumor growth and metastasis. As the role of PISD in cancer processes is not well understood, the anticancer potential of inhibitors targeting this enzyme remains uncertain.

### 4.3. Phosphatidylserine Decarboxylase Inhibitors

To date, no inhibitors have been developed for animal PISD. However, inhibitors for bacterial and yeast PISD are known. These include YU253454, YU224252, YU196325, YU253467 along with its analog YU254403 for PISD from *Candida albicans* [113], and GNF-Pf-3801/MMV007285 along with its analog 7-chloro-N-(4-ethoxyphenyl)-4-quinolinamine for PISD from *Plasmodium falciparum* (Figure 4) [114]. GNF-Pf-3801, with an IC_50_ of 4 μM, inhibits the growth of *P. falciparum*, while in human foreskin fibroblasts, this value is over eight times higher [114]. 7-chloro-N-(4-ethoxyphenyl)-4-quinolinamine demonstrated low toxicity in an in vivo mouse study and was effective against *Plasmodium falciparum* infection. Therefore, these PISD inhibitors could be used in the treatment of pathogenic yeast infections or malaria. However, PISD inhibitors have not yet been tested in clinical trials in humans.

## 5. Bioinformatics Analysis of the Role of Enzymes Interconverting PC, PE, and PS in Cancer Processes

The bioinformatics analysis was conducted using the Gene Expression Profiling Interactive Analysis (GEPIA) portal (http://gepia.cancer-pku.cn/detail.php, accessed on 1 May 2024) [100]. This portal processes raw data from The Cancer Genome Atlas (TCGA) [115]. GEPIA enables the analysis of any gene regarding its expression level in tumors compared to adjacent non-cancerous and healthy tissues, the association of gene expression levels with patient prognoses in cancer, and the correlation of expression of two genes in tumors. The significance of a gene in oncogenic processes is best reflected by the correlation between its expression level in tumors and patient prognoses across various cancer types.

In two types of cancer—lung squamous cell carcinoma and skin cutaneous melanoma—higher *PEMT* expression in the tumor is associated with a poorer prognosis. Conversely, in only one type of cancer, pancreatic adenocarcinoma, a higher expression of this gene correlates with better outcomes. This suggests that PEMT is involved in cancer progression in a subset of cancers.

For *PTDSS1*, higher expression is linked to worse prognosis in nine out of In total, thirty-three cancer types, with an additional four types showing a trend toward worse outcomes. In total, 40% of cancer types show a negative correlation between *PTDSS1* expression and prognosis. However, this pattern is not universal, as in kidney renal clear cell carcinoma, higher *PTDSS1* expression is associated with better outcomes. This indicates that PTDSS1 plays a significant role in cancer progression across many cancer types.

Similarly, higher *PTDSS2* expression is associated with poorer prognosis in four cancer types, two of which also show worse outcomes with elevated *PTDSS1* expression. In contrast, in pancreatic adenocarcinoma, higher *PTDSS2* expression is linked to a better prognosis. This suggests that PTDSS1 and PTDSS2 may complement each other with pro-tumorigenic effects in certain cancer types.

*PISD* expression shows a more variable pattern: higher expression is associated with a worse prognosis in two cancer types and a better prognosis in three. This suggests that PISD does not have a uniform role in cancer processes and may act as either a pro- or anti-tumor factor depending on the cancer type.

The main takeaway from the bioinformatic analysis is the significant role of PTDSS1 in the progression of many cancer types, making it a promising therapeutic target (Table 1).

## 6. Conclusions

Enzymes catalyzing the interconversion of PC, PE, and PS play critical physiological roles and are implicated in various diseases. Based on the collected data, the following conclusions can be drawn:-PEMT demonstrates both pro- and anticancer properties. It is also involved in conditions such as obesity, insulin resistance in obesity, HCV infection, type II diabetes, and atherosclerosis, while offering protection against NAFLD. Specific PEMT inhibitors have not yet been developed, but if they are in the future, they hold significant therapeutic potential, particularly for treating obesity. However, these inhibitors could have side effects, such as inducing NAFLD and impairing nervous system function. This latter issue could potentially be mitigated by designing PEMT inhibitors that do not cross the blood–brain barrier.-Phosphatidylserine synthases PTDSS1 and PTDSS2 play key roles in cancer progression, making them promising therapeutic targets. The inhibitors of PTDSS1 have shown anticancer potential in in vivo studies, suggesting they could be developed into effective cancer treatments. However, no clinical trials have been conducted with PTDSS1 inhibitors, meaning these compounds have not yet advanced to clinical use.-The role of PISD in disease is not well understood. The reduced activity of this enzyme in the brain may be associated with aging, highlighting the need for further research into its function in the brain and the consequences of altered activity in older individuals. Such studies could enhance our understanding of the aging process and potentially lead to strategies for slowing or mitigating its effects.

## Figures and Tables

**Figure 1 ijms-25-10745-f001:**
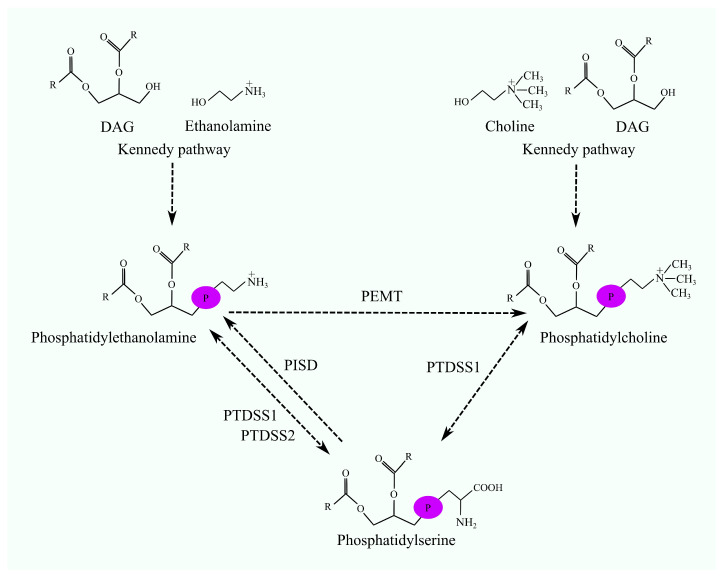
The interconversions among PC, PE, and PS. PC and PE are synthesized via the Kennedy pathway. Subsequently, these phospholipids can undergo further reactions leading to conversions into other phospholipids. Specifically, PC and PE can be enzymatically converted to PS by PTDSS1 and PTDSS2 enzymes, respectively. Moreover, PS can be transformed into PE with the involvement of PISD. Another potential reaction involves the conversion of PE to PC, catalyzed by PEMT.

**Figure 2 ijms-25-10745-f002:**
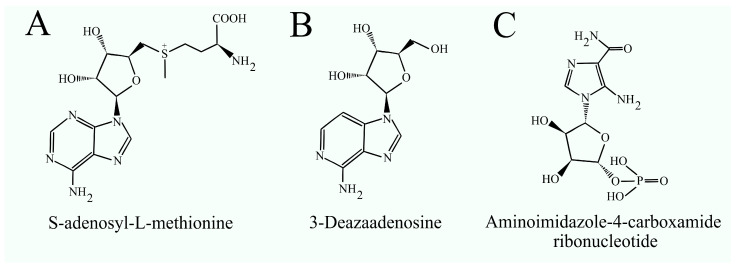
Structural formulas of (**A**) the substrate for PEMT, S-adenosyl-L-methionine, and the PEMT inhibitors (**B**) 3-deazaadenosine and (**C**) aminoimidazole-4-carboxamide ribonucleotide.

**Figure 3 ijms-25-10745-f003:**
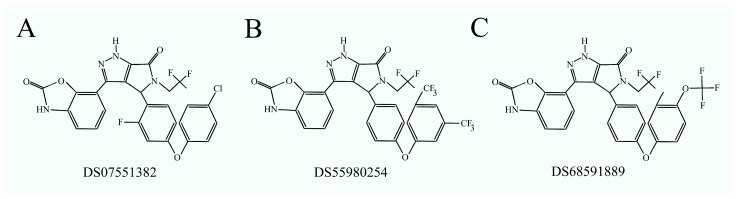
Structural formulas of PTDSS1 inhibitors: (**A**) DS07551382, (**B**) DS55980254, and (**C**) DS68591889.

**Figure 4 ijms-25-10745-f004:**
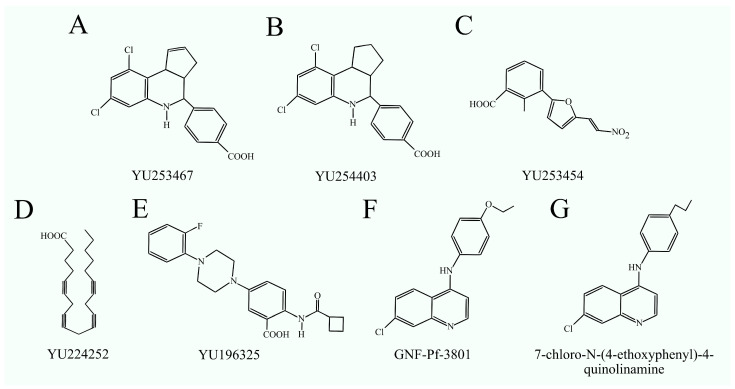
Structural formulas of PISD inhibitors: (**A**) YU253467, (**B**) YU254403, (**C**) YU253454, (**D**) YU224252, (**E**) YU196325, (**F**) GNF-Pf-3801, and (**G**) 7-chloro-N-(4-ethoxyphenyl)-4-quinolinamine.

**Table 1 ijms-25-10745-t001:** Impact of the expression levels of enzymes interconverting PC, PE, and PS on overall survival for patients with selected cancers.

Name of the Cancer	*PEMT*	*PTDSS1*	*PTDSS2*	*PISD*
Adrenocortical carcinoma	-	↓	↓ *p* = 0.10	-
Bladder urothelial carcinoma	-	-	↓	↑
Breast invasive carcinoma	-	↓ *p* = 0.088	-	-
Cervical squamous cell carcinoma and endocervical adenocarcinoma	-	-	-	-
Cholangiocarcinoma	↓ *p* = 0.085	-	-	-
Colon adenocarcinoma	-	-	-	-
Lymphoid neoplasm diffuse large B-cell lymphoma	-	-	-	-
Esophageal carcinoma	-	-	-	-
Glioblastoma multiforme	-	-	↓	-
Head and neck squamous cell carcinoma	-	↓	-	-
Kidney chromophobe	-	↓	↓ *p* = 0.088	-
Kidney renal clear cell carcinoma	-	↑	-	↓ *p* = 0.083
Kidney renal papillary cell carcinoma	-	-	↑ *p* = 0.094	-
Acute myeloid leukemia	-	-	-	-
Brain lower grade glioma	-	↓	-	-
Liver hepatocellular carcinoma	-	↓	↓	↓
Lung adenocarcinoma	-	↓ *p* = 0.072	-	-
Lung squamous cell carcinoma	↓	-	-	-
Mesothelioma	↓ *p* = 0.082	↓	-	-
Ovarian serous cystadenocarcinoma	-	↓ *p* = 0.089	-	-
Pancreatic adenocarcinoma	↑	-	↑	↑
Pheochromocytoma and Paraganglioma	-	-	-	-
Prostate adenocarcinoma	-	↓	-	-
Rectum adenocarcinoma	-	-	-	↑
Sarcoma	-	↓	-	-
Skin cutaneous melanoma	↓	↓ *p* = 0.057	-	-
Stomach adenocarcinoma	-	-	-	-
Testicular germ cell tumors	-	-	-	-
Thyroid carcinoma	-	-	-	-
Thymoma	-	↑ *p* = 0.093	↑ *p* = 0.061	-
Uterine corpus endometrial carcinoma	-	-	-	-
Uterine carcinosarcoma	-	-	-	-
Uveal Melanoma	-	↓	↓	↓

↓, red background—higher expression is associated with a worse prognosis for a patient with a given cancer; ↑, blue background—higher expression is associated with a better prognosis for a patient with a given cancer; -, gray background—expression of a particular gene is not significantly associated with patient prognosis.

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
