# Peer review of "Biochemistry and Diseases Related to the Interconversion of Phosphatidylcholine, Phosphatidylethanolamine, and Phosphatidylserine"

_ijms, 2024, doi:10.3390/ijms251910745_

Round 1

Reviewer 1 Report

Comments and Suggestions for Authors

This review provides a comprehensive description of the enzymes involved in the interconversion of Phosphatidylcholine, phosphatidylethanolamine and phosphatidylserine. The content is interesting and very informative to both the researcers in lipidomics and biomedical areas. From this paper, we know that phosphatidylserine synthases are closely linked to oncogenesis in nearly half of all cancer types, and their inhibitors' development may benefit the cancer treatment. The ovreall quality of this review is good. If authors could provide a more structurral description of these enzymes with figures illustrations, that would fulfill its content even better.

Author Response

Rev.1.

This review provides a comprehensive description of the enzymes involved in the interconversion of Phosphatidylcholine, phosphatidylethanolamine and phosphatidylserine. The content is interesting and very informative to both the researcers in lipidomics and biomedical areas. From this paper, we know that phosphatidylserine synthases are closely linked to oncogenesis in nearly half of all cancer types, and their inhibitors' development may benefit the cancer treatment. The ovreall quality of this review is good. If authors could provide a more structurral description of these enzymes with figures illustrations, that would fulfill its content even better.

The structural models of enzymes have been added.

Reviewer 2 Report

Comments and Suggestions for Authors

This review by Korbecki et al. addresses the biochemistry and disease implications of the interconversion between phosphatidylcholine (PC), phosphatidylethanolamine (PE), and phosphatidylserine (PS). The topic is of significant interest, given the critical roles these phospholipids play in cellular function and the impact of their dysregulation on diseases such as cancer, obesity, and diabetes. However, the manuscript, in its current form, is not suitable for publication due to several critical issues, including a lack of organization, focus, and clarity, as well as limited new insights into the field. Below, I outline my major concerns and provide suggestions for improvement:

1.       The introduction is underdeveloped, providing insufficient background on the roles of PC, PE, and PS, as well as the enzymes involved in their biosynthesis. Additionally, it contains only one reference. The introduction should be expanded to include a comprehensive overview of the physiological roles of these phospholipids and their biosynthetic enzymes. This section should also be well-supported with relevant citations to establish the context and importance of the review.

2.       The manuscript does not significantly advance the field, offering limited new information. The authors should aim to provide a more in-depth analysis or present novel perspectives that have not been extensively covered in previous literature. They might consider highlighting recent advancements or unresolved questions in the field to enhance the manuscript's contribution.

3.       The manuscript includes only one figure and one table, which are insufficient to convey the complexity of the topic. The review would benefit from additional figures, such as diagrams of the biosynthetic pathways and structures of key inhibitors. More tables summarizing enzyme functions, inhibitors, and disease correlations would also improve readability and comprehension.

4.       The sections discussing potential inhibitors of the enzymes are underdeveloped and lack detailed information. The authors should expand these sections by providing detailed descriptions of known inhibitors, including their chemical structures, mechanisms of action, and the current status of their development as therapeutic agents. Including comparisons between inhibitors would also be beneficial.

5.       The manuscript lacks sufficient mechanistic insights into how the enzymes involved in PC, PE, and PS interconversion contribute to disease development. The authors should include more detailed discussions on the biochemical mechanisms by which these enzymes influence disease processes. This could involve a deeper exploration of enzyme regulation, post-translational modifications, and interactions with other cellular pathways.

6.       The review focuses primarily on cases of enzyme overexpression, neglecting instances of underexpression or loss of function. A more balanced discussion should be provided, addressing both the pathological implications of enzyme overexpression and underexpression. This would offer a more comprehensive understanding of their roles in health and disease.

7.       The manuscript does not sufficiently explore the causes behind alterations in enzyme activity or expression. The authors should discuss genetic, environmental, and epigenetic factors that contribute to the dysregulation of these enzymes. This would enhance the review's depth and provide insights into potential therapeutic targets.

8.       Section 5 is poorly structured and lacks a clear conclusion. This section should be rewritten to clearly state the findings of the bioinformatics analysis and their implications. The authors should ensure that the section ends with a concise summary of the main points and their relevance to the overall review.

9.       The conclusion is brief and lacks depth. The conclusion should be expanded to summarize the key findings of the review, propose potential future research directions, and discuss the therapeutic implications of targeting the enzymes involved in phospholipid interconversion.

Comments on the Quality of English Language

The quality of the English is acceptable.

Author Response

Rev.2.

This review by Korbecki et al. addresses the biochemistry and disease implications of the interconversion between phosphatidylcholine (PC), phosphatidylethanolamine (PE), and phosphatidylserine (PS). The topic is of significant interest, given the critical roles these phospholipids play in cellular function and the impact of their dysregulation on diseases such as cancer, obesity, and diabetes. However, the manuscript, in its current form, is not suitable for publication due to several critical issues, including a lack of organization, focus, and clarity, as well as limited new insights into the field. Below, I outline my major concerns and provide suggestions for improvement:

  1. The introduction is underdeveloped, providing insufficient background on the roles of PC, PE, and PS, as well as the enzymes involved in their biosynthesis. Additionally, it contains only one reference. The introduction should be expanded to include a comprehensive overview of the physiological roles of these phospholipids and their biosynthetic enzymes. This section should also be well-supported with relevant citations to establish the context and importance of the review.

The introduction has been expanded according to the reviewer’s recommendation.

  1. The manuscript does not significantly advance the field, offering limited new information. The authors should aim to provide a more in-depth analysis or present novel perspectives that have not been extensively covered in previous literature. They might consider highlighting recent advancements or unresolved questions in the field to enhance the manuscript's contribution.

In our study, we were the first to perform a bioinformatic analysis of the significance of the expression levels of enzymes involved in phospholipid conversion in cancer processes. Higher PTDSS1 expression in the tumor is associated with poorer prognosis. Thus, we demonstrated that PTDSS1 is significant in various types of cancer, making it a potential therapeutic target. We emphasized this aspect in the revised Chapter 5.

Additionally, in our study, we showed that inhibitors of the described enzymes have not been developed (in the case of PEMT and PISD) or have been developed (in the case of PTDSS1), and therefore, the therapeutic value of these compounds has not yet been explored.

  1. The manuscript includes only one figure and one table, which are insufficient to convey the complexity of the topic. The review would benefit from additional figures, such as diagrams of the biosynthetic pathways and structures of key inhibitors. More tables summarizing enzyme functions, inhibitors, and disease correlations would also improve readability and comprehension.

Two figures depicting the structural models of the inhibitors of the described enzymes have been added.

  1. The sections discussing potential inhibitors of the enzymes are underdeveloped and lack detailed information. The authors should expand these sections by providing detailed descriptions of known inhibitors, including their chemical structures, mechanisms of action, and the current status of their development as therapeutic agents. Including comparisons between inhibitors would also be beneficial.

A more detailed description of the inhibitors has been added.

  1. The manuscript lacks sufficient mechanistic insights into how the enzymes involved in PC, PE, and PS interconversion contribute to disease development. The authors should include more detailed discussions on the biochemical mechanisms by which these enzymes influence disease processes. This could involve a deeper exploration of enzyme regulation, post-translational modifications, and interactions with other cellular pathways.

We agree with the reviewer that changes in the activity of the described enzymes do not directly cause diseases. It is the alterations in phospholipid levels that are the direct disease factor. Therefore, we have added brief sections discussing the connection between changes in phospholipid levels and diseases, linking enzyme activity changes to disease development.

  1. The review focuses primarily on cases of enzyme overexpression, neglecting instances of underexpression or loss of function. A more balanced discussion should be provided, addressing both the pathological implications of enzyme overexpression and underexpression. This would offer a more comprehensive understanding of their roles in health and disease.

Typically, under the influence of various diseases, the expression of the enzymes we discuss increases. These enzymes also participate in disease mechanisms, and therefore, the use of their inhibitors would theoretically have a therapeutic effect. The imbalance in the discussion, as noted by the reviewer, where overexpression is addressed more often than reduced expression, stems from the behavior of the enzymes in question.

  1. 7. The manuscript does not sufficiently explore the causes behind alterations in enzyme activity or expression. The authors should discuss genetic, environmental, and epigenetic factors that contribute to the dysregulation of these enzymes. This would enhance the review's depth and provide insights into potential therapeutic targets.

In the available experimental publications, the causes of changes in the expression of the enzymes we describe under the influence of diseases have not been studied. As a result, there is a lack of knowledge regarding the mechanisms behind the changes in the expression of these enzymes in response to disease processes.

  1. Section 5 is poorly structured and lacks a clear conclusion. This section should be rewritten to clearly state the findings of the bioinformatics analysis and their implications. The authors should ensure that the section ends with a concise summary of the main points and their relevance to the overall review.

The section has been rewritten, and the main conclusion of the bioinformatic analysis has been added.

  1. The conclusion is brief and lacks depth. The conclusion should be expanded to summarize the key findings of the review, propose potential future research directions, and discuss the therapeutic implications of targeting the enzymes involved in phospholipid interconversion.

The conclusions have been expanded, with an emphasis on the potential of inhibitors of the described enzymes and directions for future research.

Reviewer 3 Report

Comments and Suggestions for Authors

The authors provide a very complete review enumerating literature evidence of the role of enzymes enables conversion of phospholipids into others on a wide variety of diseases. The review is detailed and can be used as a first step to find works dealing with this topic and a given type of disease. What I miss and the authors could maybe add as example for a given disease, is the microscopic view of how excess or defect of a given phospholipid affects or induces a given disease and provide relevant references. 

In addition, the authors could add one or two figures or sketches more for a particular case, to make the review easily readable and less heavy

Comments on the Quality of English Language

minor changes required after grammar crosscheck

Author Response

Rev. 3.

The authors provide a very complete review enumerating literature evidence of the role of enzymes enables conversion of phospholipids into others on a wide variety of diseases. The review is detailed and can be used as a first step to find works dealing with this topic and a given type of disease. What I miss and the authors could maybe add as example for a given disease, is the microscopic view of how excess or defect of a given phospholipid affects or induces a given disease and provide relevant references.

We agree with the reviewer that changes in the activity of the described enzymes do not directly influence or cause diseases. It is the alterations in phospholipid levels that are the direct disease factor. Therefore, we have added brief sections discussing the connection between changes in phospholipid levels and diseases, linking enzyme activity changes to disease development.

In addition, the authors could add one or two figures or sketches more for a particular case, to make the review easily readable and less heavy

Two figures have been added.

Round 2

Reviewer 2 Report

Comments and Suggestions for Authors

The authors have addressed most of my concerns by expanding the introduction, adding figures, providing more detailed discussions of inhibitors, and improving the structure and depth of sections. In my opinion, the manuscript can be accepted for publication in its current form. 

Comments on the Quality of English Language

The quality of the English language is acceptable.